# Feasibility of Efficient, Direct, Butanol Production from Food Waste without Nutrient Supplement by *Clostridium saccharoperbutylacetonicum* N1-4

Xiaona Wang [1,†], Haishu Sun [1,†], Yonglin Wang [1], Fangxia Wang [1], Wenbin Zhu [1], Chuanfu Wu [1,2], Qunhui Wang [1,2] and Ming Gao [1,2,*]

1    Department of Environmental Engineering, School of Energy and Environmental Engineering, University of Science and Technology Beijing, 30 Xueyuan Road, Haidian District, Beijing 100083, China
2    Beijing Key Laboratory on Resource-Oriented Treatment of Industrial Pollutants, University of Science and Technology Beijing, 30 Xueyuan Road, Haidian District, Beijing 100083, China
*    Correspondence: gaoming402@gmail.com; Tel./Fax: +86-010-62332778
†    These authors contributed equally to this work.

**Abstract:** This study investigated the feasibility of direct butanol production from starchy food waste (without saccharification and nutrient supplementation). First, *Clostridium saccharoperbutylacetonicum* N1-4 was selected as an efficient starch-utilizing clostridia, and amylose was used by the strain more readily than amylopectin for solvent production. Furthermore, direct fermentation avoided substrate inhibition due to saccharification and produced 12.1 g/L of butanol at a production rate of 0.705 g/L/h and a yield of 0.402 C-mol/C-mol with a solid–liquid ratio of 1:1 (*w/v*). At a solid–liquid ratio of 1:2 (*w/v*), the maximum butanol production rate in the direct mode was 2.05 times higher than that in the saccharified mode. Elemental analysis demonstrated that the food waste analyzed was rich in trace elements and, hence, exogenous nutrient supplementation was unnecessary. Collectively, direct butanol production from food waste could function as a low-cost, highly efficient, and simple fermentative process, which is a promising strategy for food waste disposal.

**Keywords:** butanol production; food waste; direct fermentation; no nutrient supplementation; *Clostridium saccharoperbutylacetonicum*

## 1. Introduction

With the increasing demand for energy, industrialization, and mobility in global development, the energy required for powering engines has also dramatically increased, leading to the significant consumption of non-renewable natural resources and fossil fuels. At present, 80% of global energy consumption comes from fossil fuels, of which 58% are used only for transportation [1]. While fossil fuels are a highly efficient global energy source, greenhouse gas emissions are one of the root causes of greenhouse effects, biodiversity reduction, and the increase in sea level. Therefore, the development of renewable biomass energy is urgently required [2].

Butanol is a potential new biofuel that is superior to traditional biofuel ethanol, as it not only yields a higher amount of energy (energy density: butanol: 30 MJ/L, ethanol: 19 MJ/L) [3], but also exhibits lower volatility and corrosiveness. Furthermore, butanol can be mixed with gasoline in any proportion or used as a fuel independently [4]. The industrial production of traditional biomass butanol mainly utilizes starchy grains or agricultural by-products as raw materials [5]. Following hydrolysis, a mixture of acetone, butanol, and ethanol is obtained by the anaerobic fermentation of *Clostridium*, and the corresponding products are obtained by distillation. Therefore, butanol fermentation is also called acetone–butanol–ethanol (ABE) fermentation [6]. Numerous studies have focused on using lignocellulosic and starchy biomass as substrates for butanol production. However,

strict pre-treatment conditions are required for lignocellulosic biomass utilization, which leads to low substrate conversion rates, high energy consumption, and high production costs. Toxic by-products, such as syringaldehyde, furfural, 5-HMF, and *p*-coumaric acid, are generated during lignocellulose pre-treatment, which are harmful to ABE fermentation [4,7]. Starchy substrates can be directly fermented and utilized by ABE-producing clostridia without pre-treatment or enzymolysis [8]. However, most starchy substrates are derived from grain crops, which increases the production costs. At present, bio-butanol produced via ABE fermentation lacks commercial competitiveness because of its low production rate and high feedstock cost [9]. Raw feedstock accounts for up to 60% of the overall bio-butanol production cost [10]. Thus, the search for low-cost, alternative, non-edible biomass and improved production efficiency are essential to future developments [11].

With the increase in the urban population and improvement to people's quality of life, the production of food waste (FW) is also increasing, resulting in increasingly severe environmental problems. FW is characterized by a high organic content, which also comprises various nutrient elements (proteins, fatty acids, and minerals) for microbial growth [12]. Numerous studies have utilized FW as a raw substrate to produce high value-added products, such as lactic acid, methane, microbial lipid, and bio-ethanol [13,14]. However, the use of FW as a substrate for butanol production has rarely been reported. Huang et al. [15] and Ujor et al. [16] conducted butanol production from simulated FW and industrial starchy food residues by *C. beijerinckii*; however, exogenous P2 stock was supplied as a nitrogen source, trace mineral, and pH buffer. At present, no study has investigated the feasibility of butanol production from real FW without the addition of exogenous nutrition. *Clostridium saccharoperbutylacetonicum* is one of the major species of ABE-producing clostridia, and Thang et al. [17] previously reported direct butanol production using *C. saccharoperbutylacetonicum* N1-4 from cassava and sago starches. However, there are few reported studies on direct butanol production from FW using *C. saccharoperbutylacetonicum*.

The aim of this study is to investigate the feasibility of using FW as a substrate for butanol production without enzymatic saccharification and exogenous nutrition supplementation. The most efficient starch utilization strain is selected among the three mainstream ABE-producing bacteria: *C. acetobutylicum*, *C. beijerinckii*, and *C. saccharoperbutylacetonicum*. The proposed direct fermentation strategy is evaluated not only in terms of starch composition, substrate inhibition concentration, kinetic parameters, and trace nutrition elements, but also the superiority of saccharification in terms of butanol production and carbon conversion rates without exogenous nutrition. This study is the first to report the direct production of butanol by the fermentation of FW with *C. saccharoperbutylacetonicum* without nutrient supplementation, which provides a scientific basis for butanol production by the direct fermentation of starchy, organic waste.

## 2. Materials and Methods

### 2.1. Food Waste

The FW used in this study was collected from the student canteen at the University of Science and Technology Beijing. The FW was primarily composed of discarded rice, pasta, meat, vegetables, peels, bones, and eggshells. Large bones, plastic, and paper towels were removed. The remaining waste was milled with a meat grinder and stored at $-20\ ^\circ\text{C}$. The frozen FW was thawed for 8–10 h at room temperature prior to use. The composition of the collected canteen FW was analyzed and is presented in Table 1. The composition measurement procedures are provided in the analysis section.

**Table 1.** Canteen food waste (FW) composition.

| Parameters | TS [a] | TN [b,d] | TOC [c,d] | Starch [d] | Glucose [d] | pH [e] |
|---|---|---|---|---|---|---|
| FW (%) | 26.5 | 2.16 | 48.3 | 61.7 | 4.03 | 4.84 |

[a] Total solids ($w/w$). [b] Total nitrogen ($w/w$). [c] Total organic carbon ($w/w$). [d] The parameters are calculated for dry matter ($w/w$). [e] The natural pH value of FW medium without adjustment.

## 2.2. Microorganisms and Media

*Clostridium saccharoperbutylacetonicum* N1-4 (ATCC 13564), *C. acetobutylicum* ATCC 824, and *C. beijerinckii* NCIMB 8052 were used in this study. Potato glucose culture medium (PG) was used to preserve and activate the spores of these strains. The strains were kept at 4 °C, as were the spores in the PG medium. One milliliter of spore suspension was transferred aseptically to 9 mL of fresh PG medium (10%, *v/v*), heat-stocked (*C. saccharoperbutylacetonicum* N1-4, boiling water for 1 min; *Clostridium acetobutylicum* ATCC 824 and *C. beijerinckii* NCIMB 8052, 80 °C water for 10 min), cultured for 24–28 h (*C. saccharoperbutylacetonicum* N1-4, at 30 °C; *C. acetobutylicum* ATCC 824 and *C. beijerinckii* NCIMB 8052, at 37 °C) in an AnaeroBox (Mitsubishi Gas Chemical America, Inc., New York, NY, USA), and used as an inoculum. Tryptone-yeast extract-acetate (TYA) medium was used for the pre- and main cultures with the following composition per liter of deionized water: 20–80 g glucose/starch, 2 g yeast extract (Aladdin Reagent, Shanghai, China), 6 g tryptone (Aladdin Reagent, Shanghai, China), 3 g $CH_3COONH_4$, 0.3 g $MgSO_4 \cdot 7H_2O$, 0.5 g $KH_2PO_4$, and 10 mg $FeSO_4 \cdot 7H_2O$. The initial pH of the TYA medium was adjusted to 6.5 using 3 M KOH. In all the experiments, the carbon source (glucose and starch) and other components were sterilized separately at 115 °C for 15 min, and then mixed aseptically. The FW medium consisted of FW and a deionized water mixture with a solid–liquid ratio of 1:1 or 1:2 (*w/v*). Different saccharification processes observed with the addition of two commercial enzymes in the FW medium, separately, are described in detail in the subsequent section. The initial pH was adjusted to 6.2–6.5. The FW medium was sterilized at 121 °C for 15 min.

## 2.3. Culture Conditions and Acetone–Butanol–Ethanol Fermentation

The pre-culture of ABE-producing clostridia was anaerobically performed in TYA medium containing 20 g/L glucose for 15–17 h. The main cultures were inoculated with 10% (*v/v*) of pre-culture (the initial dry cell weight was around 0.20 g/L), without pH control, under anaerobic conditions by sparging with filtered (Dismisc-25HP, 0.22 μm, Advantec, Tokyo, Japan) oxygen-free nitrogen gas for 15 min. Different types of ABE-producing clostridia were cultured at appropriate temperatures, as described in the previous section.

To compare the starch-utilizing capacity of different clostridia during ABE fermentation, a series of batch cultures were performed in 20 mL test tubes with a 10 mL working volume of TYA medium, containing 20 g/L of starch. Furthermore, to investigate the effect of the amylose/amylopectin ratio in starch from different sources on solvent production, pure amylose (amylose 100% [*w/w*]; Aladdin Reagent), corn starch (amylose 27% [*w/w*], amylopectin 73% [*w/w*]; Aladdin Reagent), and potato starch (amylose 19% [*w/w*], amylopectin 81% [*w/w*]; Aladdin Reagent) were used as the sole carbon sources in individual TYA mediums. Following strict anaerobic cultivation for 72 h in AnaeroBox, the culture broths were collected and analyzed.

Batch fermentation in the FW medium was performed in a 500 mL flask (300 mL working volume) with silicone rubber stoppers. To evaluate the feasibility of direct butanol production from the FW and to compare the direct fermentation mode with saccharified one, four groups of experiments were performed with different enzymatic saccharification strategies.

One sample included FW medium saccharified by α-amylase (FW-A). First, enzymatic hydrolysis was performed at 75 °C for 2 h under a pH of 6.5 by adding 0.67 μL/g FW (wet weight) α-amylase (527.24 KNU/g [KNU; kilo novo unit]; Br, Aladdin Reagent). An additional FW medium was saccharified by α-amylase and glucoamylase (FW-AG). After performing α-amylase hydrolysis, as previously described, saccharification proceeded further at 60 °C for 6 h by adding 1.33 μL/g FW (wet weight) of glucoamylase ($10^5$ U/mL; from *Aspergillus*, Aladdin Reagent) at a pH of 4.5 (adjusted by using 3 M HCl). A third sample of FW medium was only saccharified by glucoamylase (FW-G), and glucoamylase hydrolysis conditions were consistent with those previously stated. The FW medium used in the direct fermentation experiment (FW-D) was without saccharification. Before sterilization, the pH values of the FW mediums were adjusted to 6.2–6.5. Butanol production

from FW in the four groups was initiated by inoculating 10% (*v/v*) pre-culture of the N1-4 strain and cultivated at 30 °C for 72 h. The samples were withdrawn from the fermentation broths at various time points for glucose, starch, and product measurements. Figure 1 is the flowchart of this experiment.

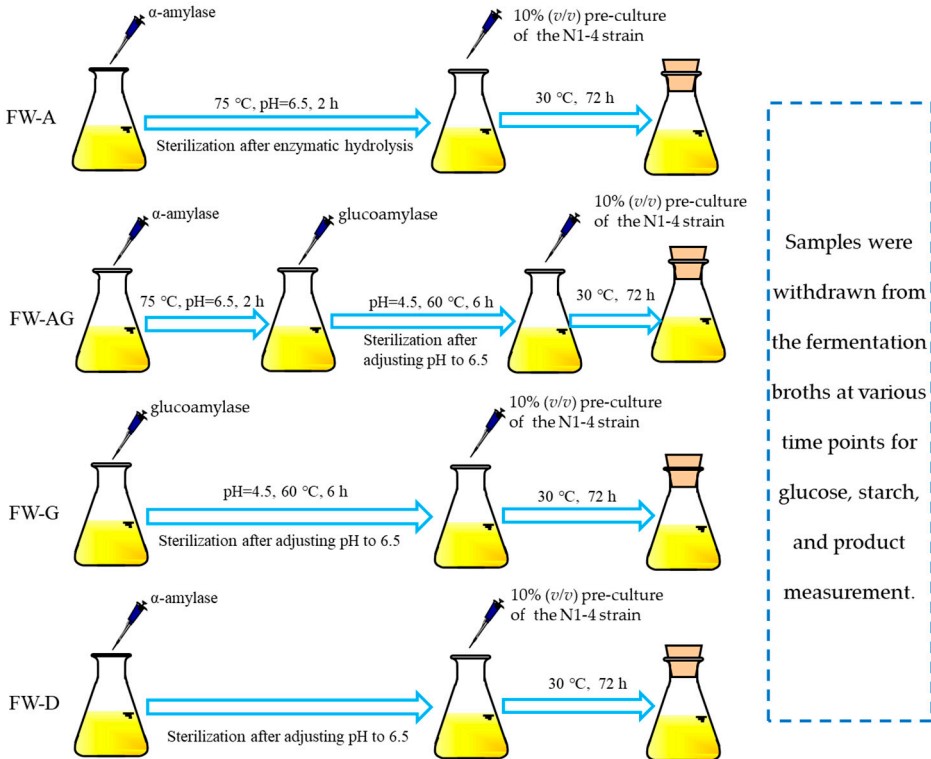

**Figure 1.** Experimental flowchart.

To investigate the effect of the initial substrate concentration on butanol production, batch cultures were performed in 500 mL flasks containing 300 mL of TYA medium supplemented with different concentrations of glucose or starch ranging from 40 to 80 g/L. The samples were withdrawn periodically, over a period of 72 h of cultivation, for the analysis of the products and substrates.

### 2.4. Analytical Procedures

The total solid (TS) content of the FW was determined using the gravimetric method [15]. Dry FW was ground and sieved (mesh size: 0.25 mm × 0.25 mm) into fine powder for total organic carbon (TOC), total nitrogen (TN), and trace mineral measurements. TOC and TN were determined using an elemental analyzer (Vario EL cube, Elementar, Germany) with complete combustion at 1050 °C. The trace mineral composition of the FW was measured using inductively coupled plasma optical emission spectroscopy (ICP-OES) (Optima 8000, PerkinElmer Inc., Waltham, MA, USA). The pH values of the fermented samples were determined using a pH meter (PHS-3E; Leici, Shanghai, China).

The starch concentration was determined by the double-enzymatic method and modified according to the method of Thang and Kobayashi [17]. A portion of 10 μL of α-amylase (527.24 KNU/g; Br, Aladdin Reagent) and 990 μL of 50 mM 3-(N-morpholino)-propane-sulfonic acid (MOPS) (pH 7.0) buffer solution was added to 1 mL of culture medium and incubated at 75 °C for 2 h to hydrolyze the starch in the medium to soluble dextrin. Subsequently, 980 μL of 0.1 M sodium acetate buffer (pH 4.5) and 20 μL of gluczyme ($10^5$ U/mL; from *Aspergillus*, Aladdin Reagent) were added to the solution and then incubated at 60 °C for 6 h to hydrolyze dextrin in the medium to glucose. The solution was allowed to cool down to room temperature and then centrifuged at 14,000× *g* for 10 min at 4 °C, and the

supernatant was filtered through a membrane (Nylon66-Φ13, 0.45 μm, Jinteng, Tianjin, China). The glucose concentration in this solution was determined using high-performance liquid chromatography (HPLC). The starch concentration in the fermentation broth was calculated as follows [16]:

$$\text{Starch concentration (g/L)} = \text{glucose concentration (g/L)} \times df \times hf$$

where *df* is dilution factor 3 and *hf* is the starch hydrolysis factor of 0.91.

Glucose, lactic acid, and acetic acid concentrations in the fermentation broth were detected using HPLC (LC-20A; Shimadzu, Kyoto, Japan) equipped with an SH 1011 column (Shodex, Tokyo, Japan). The concentrations of ABE solvents and butyric acid were measured using a gas chromatograph (GC2010Plus; Shimadzu, Kyoto, Japan) equipped with a flame ionization detector (FID) and a 30 m capillary column (DB-FFAP; i.d. 0.53 mm; 125-3237; Agilent Technologies, Santa Clara, CA, USA) at a split ratio of 25:1.

*2.5. Calculation*

The conversion rate of butanol produced by the substrate carbon source was calculated as follows: 1 g of starch was hydrolyzed to 1.1 g of glucose [18]:

$$\text{Butanol yield (C-mol/C-mol)} = (C_B/M_B \times 4)/((C_G + 1.1 \times C_S)/M_G \times 6)$$

where $C_B$ is butanol production (g/L), $M_B$ is butanol molar mass (74.12 g/mol), $C_G$ is glucose consumption (g/L), $C_S$ is starch consumption (g/L), and $M_G$ is the molar mass of glucose (180.16 g/mol).

**3. Results and Discussion**

*3.1. Direct Butanol Production from Starch by Different Aceton–Butanol–Ethanol-Producing Clostridia*

Although the composition of real FW is complex, starch is considered the primary carbon source component in FW according to human consumption habits in China [19]. In this study, the starch content of the FW substrate reached 61.7% (Table 1).

To select a suitable ABE-producing strain, this study examined the ability of three ABE-producing clostridia (*C. saccharoperbutylacetonicum*, *C. acetobutylicum*, and *C. beijerinckii*) in order to directly utilize starch. To ensure the experimental results, the inoculum cell concentrations of the three strains were all diluted and adjusted to approximately 0.20 g/L for the dry cell weight of the inoculum. The three strains were inoculated into TYA medium containing 20 g/L of starch for 72 h. The results are presented in Table 2. The production of butanol by strain N1-4 using pure amylose, corn starch, and potato starch was superior to that of the other two clostridia species. Particularly, butanol and total solvent yields using pure amylose were the highest (butanol 6.04 g/L; total ABE 7.21 g/L): 1.51 and 1.33 times that of *C. acetobutylicum* ATCC 824 and 2.30 and 1.95 times that of *C. beijerinckii* NCIMB 8052, respectively. *Clostridium saccharoperbutylacetonicum* N1-4 not only utilizes starch, but also utilizes 50 g/L of glucose in the TYA medium better than *C. acetobutylicum* and *C. beijerinckii* (Figure S1, Table S1). In addition, strain N1-4 can also be used for efficient ABE fermentation using various substrates, such as cellobiose, xylose [20], and organic acids [21]. Studies have indicated that *C. saccharoperbutylacetonicum* N1-4 can directly use starch for ABE fermentation [22] because it can secrete high levels of active amylase and glucoamylase with high activity. Moreover, the rate of glucose production by enzymatic hydrolysis was faster than that of self-cell growth and the glucose used to produce the solvent. Simultaneously, it can effectively reduce the viscosity of the starch medium in the early stages of fermentation.

**Table 2.** Solvent production from starchy substrates by different ABE-producing clostridia.

| Strains | Starchy Substrate [a] | Acetone | Max. Solvents Production (g/L) | | |
| --- | --- | --- | --- | --- | --- |
| | | | Ethanol | Butanol | Total ABE |
| *C. saccharoperbutylacetonicum* N1-4 | Amylose starch | $0.778 \pm 0.057$ | $0.385 \pm 0.023$ | $6.04 \pm 0.23$ | $7.21 \pm 0.34$ |
| | Corn starch | $0.799 \pm 0.025$ | $0.479 \pm 0.015$ | $5.60 \pm 0.27$ | $6.88 \pm 0.27$ |
| | Potato starch | $0.714 \pm 0.032$ | $0.259 \pm 0.009$ | $4.99 \pm 0.18$ | $5.96 \pm 0.29$ |
| *C. acetobutylicum* ATCC 824 | Amylose starch | $1.220 \pm 0.078$ | $0.208 \pm 0.007$ | $3.99 \pm 0.19$ | $5.41 \pm 0.19$ |
| | Corn starch | $0.752 \pm 0.053$ | $0.168 \pm 0.009$ | $3.37 \pm 0.20$ | $4.29 \pm 0.21$ |
| | Potato starch | $1.070 \pm 0.038$ | $0.163 \pm 0.011$ | $2.89 \pm 0.13$ | $4.13 \pm 0.16$ |
| *C. beijerinckii* NCIMB 8052 | Amylose starch | $0.873 \pm 0.044$ | $0.186 \pm 0.011$ | $2.63 \pm 0.10$ | $3.69 \pm 0.12$ |
| | Corn starch | $0.897 \pm 0.051$ | $0.187 \pm 0.008$ | $2.46 \pm 0.10$ | $3.54 \pm 0.15$ |
| | Potato starch | $0.993 \pm 0.039$ | $0.184 \pm 0.009$ | $2.42 \pm 0.14$ | $3.60 \pm 0.20$ |

Batch cultures were performed using *C. saccharoperbutylacetonicum* N1-4 (30 °C), *C. acetobutylicum* ATCC 824 (37 °C), and *C. beijerinckii* NCIMB 8052 (37 °C), respectively, for 72 h in TYA medium containing 20 g/L of starch (working volume, 10 mL). [a] Amylose starch (pure amylose 100% (*w/w*)); corn starch (amylose 27% (*w/w*), amylopectin 73% (*w/w*)); potato starch (amylose 19% (*w/w*), amylopectin 81% (*w/w*)).

In this study, we investigated the effects of different sources and compositions of starch (pure amylose, corn starch, and potato starch) on ABE batch fermentation. In this experiment, pure amylose was composed of 100% (*w/w*) amylose, corn starch containing 27% (*w/w*) amylose and 73% (*w/w*) amylopectin, and potato starch containing 19% (*w/w*) amylose and 91% (*w/w*) amylopectin. As shown in Table 2, the ability of the three strains to use pure amylose was significantly better than that of the corn and potato starches ($p = 0.017 < 0.05$), and the production of butanol and total solvent was lower as the amylopectin component increased. For example, the production of butanol and total solvent using amylose by strain N1-4 was 1.08 and 1.05 times that of corn starch and 1.21 and 1.20 times that of potato starch, respectively. The underlying mechanism that caused the lower production of butanol and total solvent using amylopectin was that starch granules absorb water molecules and then expand and break, destroying their microcrystalline structure to form gels under heating conditions, which is called gelatinization [23]. However, following gelatinization, the chains of starch molecules are reoriented in the horizontal direction as those of the hydrogen bonding, forming the hybrid microcrystalline bundles again. This process is called retrogradation [24]. The higher the amylose content, the more retrogradation is likely to occur, that is, the solution becomes opaque and is accompanied by a white precipitate [25]. In this study, gelatinization occurred when the starch substrate was in the process of high (Figure S2), moist, heat sterilization. With a decrease in the temperature, pure amylose presented retrogradation. The formation of a transparent gel in the corn and potato starches was due to the gelatinization of amylopectin, which resulted in limited contact surfaces and affected mass transfer. Consequently, the fermentation production rate was low. On the contrary, the amylose group was not affected by retrogradation, and the best effect of butanol production was presented by fermentation. These results show that *C. saccharoperbutylacetonicum* N1-4 is more suitable for ABE fermentation directly using starch substrates; therefore, this strain was used as a fermentation strain in subsequent studies.

*3.2. Butanol Production from Food Waste with or without (Direct) Saccharification*

FW consists of a large quantity of cereal starch, which usually contains approximately 20% (*w/w*) amylose and 80% (*w/w*) amylopectin [26]. However, FW contains components in addition to starch. Therefore, we clarified whether it can be used as a substrate of *C. saccharoperbutylacetonicum* N1-4 for the production of butanol instead of starch directly. In this study, we used real FW as a raw material to compare the effect of fermentation following enzymatic saccharification with that of direct fermentation without the addition

of exogenous nutrients. The effect of different enzymatic hydrolysis conditions on the saccharification of FW was also investigated. Due to starch gelatinization at high temperatures, the gelatinization of a high amylopectin content can inhibit the contact and mass transfer between strains and the substrate in the fermentation system. Therefore, the FW was adjusted to a solid–liquid ratio of 1:1 (*w/v*) by adding water prior to fermentation. The ground waste slurry was passed through a 1.0 mm sieve and used as the raw material for fermentation. Four sets of raw waste material exposed to enzymolysis saccharification conditions were set: FW was treated with $\alpha$-amylase for 2 h (abbreviated as the FW-A group), with glucoamylase for 6 h (abbreviated as the FW-G group), and with $\alpha$-amylase for 2 h and subsequently with glucoamylase for 6 h (abbreviated as the FW-AG group), and the final group underwent direct fermentation (without enzymatic hydrolysis; referred to as the FW-D group). In each group, strains N1-4 were inoculated with 10% (*v/v*) and fermented at 30 °C for 72 h. The consumption of substrate (glucose, starch), ABE fermentation kinetic parameters, and solvent production were compared. The solvent production results for the four groups in the fermentation experiments are presented in Table 3.

Following different enzymolysis treatment methods, the initial substrates of the four groups were significantly different ($p = 0.0003 < 0.05$). Glucose levels in the FW-A group were higher than those in the FW-D group; however, the starch content was lower than that in the FW-D group. The glucose levels in the FW-G and FW-AG groups were much higher than those in the FW-A and FW-D groups; however, starch was not detected. During the fermentation of inoculated strain N1-4 in the abovementioned four groups of FW substrates, the starch and glucose in the FW-A and FW-D groups were simultaneously consumed by the strains, and the concentration was reduced. Butanol and total solvent concentrations gradually increased. The initial pH value of the two groups was 6.20, which decreased first and then increased with fermentation. FW-A reached a minimum pH of 4.97 at 36 h, while FW-D reached a minimum pH of 5.01 at 12 h (Figure 2a,d). As shown in Table 3, following fermentation for 72 h, 52.2 g/L (FW-A) and 44.7 g/L (FW-D) of substrates were consumed. The total solvent production for the FW-A group was 17.8 g/L, including 11.2 g/L of butanol. The butanol yield was 0.337 C-mol/C-mol during the entire process, and the highest butanol production rate was 0.390 g/L/h at 36 h of fermentation. In contrast, the production values of total solvent (16.1 g/L) and butanol (12.1 g/L) were equivalent to the FW-D group; however, the butanol yield in the FW-D group was even higher, reaching 0.402 C-mol/C-mol. The highest rate of butanol production was 0.705 g/L/h following 12 h of fermentation.

In addition, the starch-free FW-G and FW-AG groups exhibited almost no glucose consumption (5.06 g/L and 4.34 g/L), and the total solvent production values were 0.708 and 0.847 g/L following 72 h of fermentation, respectively (Table 3). The pH slowly decreased and did not rebound (Figure 2b,c), indicating that the fermentation of strain N1-4 was significantly inhibited. This may be because the initial glucose concentrations of the two fermentation groups were 108 and 103 g/L, respectively, exceeding 100 g/L. Substrate inhibition occurred when the substrate concentration was too high, resulting in very low butanol production. In this study, the maximum butanol production, production rate, and butanol yield were obtained in the FW-D group (Table 3), indicating that it is feasible for strain N1-4 to produce butanol by directly utilizing starch substrates in FW. Direct fermentation not only prevents the use of enzyme preparations, reducing costs, but also reduces the fermentation step and avoids inhibiting the substrate with a high glucose concentration [5].

**Table 3.** Solvents produced from food waste by strains N1-4 with or without (direct) saccharification processes.

| Group [a] | S/L [b] (*w/v*) | Max. Solvents Production (g/L) | | | | Initial Carbon Source (g/L) | | Substrates Consumption [c] (g/L) | Butanol Yield (C-mol/C-mol) | Max. Butanol Production Rate [d] (g/L/h) |
|---|---|---|---|---|---|---|---|---|---|---|
| | | Acetone | Butanol | Ethanol | Total ABE | Glucose | Starch | | | |
| FW-A | 1:1 | 5.79 ± 0.39 | 11.2 ± 0.4 | 0.760 ± 0.035 | 17.8 ± 0.7 | 30.1 ± 1.5 | 76.6 ± 4.4 | 52.2 ± 2.1 | 0.330 ± 0.010 | 0.390 ± 0.016 (36−48 h) |
| FW-G | | 0.116 ± 0.005 | 0.362 ± 0.015 | 0.230 ± 0.011 | 0.708 ± 0.037 | 108 ± 4 | N.D. [e] | 5.06 ± 0.20 | 0.116 ± 0.006 | 0.026 ± 0.001 (48−60 h) |
| FW-AG | | 0.161 ± 0.007 | 0.399 ± 0.013 | 0.287 ± 0.010 | 0.847 ± 0.028 | 103 ± 2 | N.D. [e] | 4.34 ± 0.09 | 0.149 ± 0.004 | 0.030 ± 0.001 (48−60 h) |
| FW-D | | 3.58 ± 0.17 | 12.1 ± 0.5 | 0.453 ± 0.009 | 16.1 ± 1.0 | 4.26 ± 0.22 | 92.5 ± 3.8 | 44.7 ± 1.9 | 0.402 ± 0.012 | 0.705 ± 0.015 (12−18 h) |
| FW-G | 1:2 | 2.37 ± 0.11 | 9.72 ± 0.49 | 0.063 ± 0.005 | 12.2 ± 0.6 | 54.3 ± 2.0 | N.D. [e] | 46.3 ± 2.3 | 0.340 ± 0.015 | 0.250 ± 0.009 (12−18 h) |
| FW-D | | 1.83 ± 0.09 | 9.72 ± 0.39 | 0.292 ± 0.012 | 11.8 ± 0.6 | 4.88 ± 0.12 | 46.9 ± 2.3 | 39.9 ± 1.9 | 0.363 ± 0.007 | 0.512 ± 0.015 (9−12 h) |

Each batch fermentation was independently performed using *C. saccharoperbutylacetonicum* N1-4 at 30 °C for 72 h in food waste (FW) medium with the indicated enzymatic process (working volume: 300 mL). [a] FW-A, FW hydrolyzed by amylase; FW-G, FW hydrolyzed by glucoamylase; FW-AG, FW hydrolyzed by both amylose and glucoamylase; FW-D, FW without hydrolysis. [b] S/L (*w/v*): the ratio of solid (food waste) to liquid (deionized water). [c] Total substrate consumption = glucose consumption + starch consumption. [d] Butanol production rate (g/L/h) = $(C_2 − C_1)/(t_2 − t_1)$, where $C$ is the butanol concentration (g/L), $t$ is the sampling time (h); the period for the calculation is shown in the following parenthesis. [e] N.D.: not detected by high-performance liquid chromatography (HPLC).

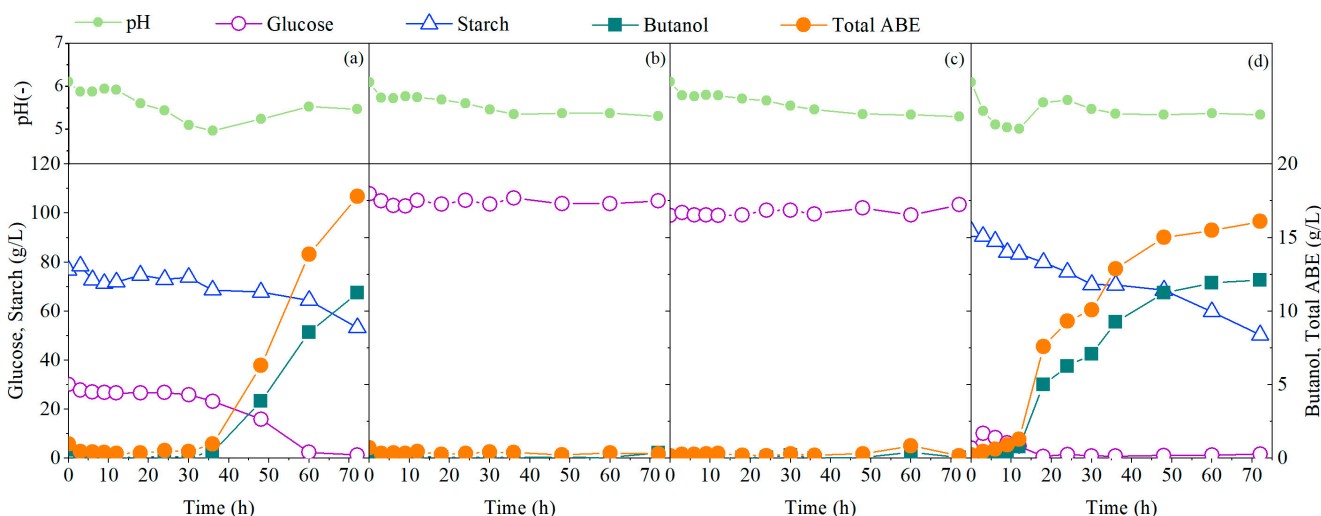

**Figure 2.** Time course of solvent production from food waste with different saccharification treatments (solid–liquid ratio of 1:1 (*w/v*)). (**a**) Food waste is treated with α-amylase for only 2 h (abbreviated as FW-A group); (**b**) food waste is treated with glucoamylase for 6 h (abbreviated as FW-G); (**c**) food waste is treated with α-amylase for 2 h and then treated with glucoamylase 6 h (abbreviated as FW-AG group); (**d**) direct ABE fermentation (without enzymatic hydrolysis; referred to as FW-D group).

It was also observed that the saccharification effect of the FW-AG group by the double enzymatic method was almost equivalent to that of the FW-G group with a single glucoamylase. This was because the two processes of gelatinization and saccharification were included in the double enzymatic method [27]. The gelatinization process hydrolyzed the α-1,4 glycosidic bond in starch with α-amylase to produce small-molecule dextrin. Then, the α-1,6 glycosidic and α-1,4 glycosidic bonds in the oligosaccharides were cut by glucoamylase, and finally, glucose was generated to complete the saccharification process [28]. In this study, FW was gelatinized during the cooking process, which meant that the heating process collapsed the raw starch micelle structure, and the starch molecules formed small-molecule dextrins [29]. Therefore, in the following experiment, only glucoamylase was added to achieve the complete saccharification effect of FW. When the solid–liquid ratio was 1:1 (*w/v*), a large amount of substrate remained after fermentation was not utilized by the bacteria, resulting in a waste of resources. Simultaneously, due to substrate inhibition, the effects of direct and saccharified fermentation processes could not be compared. Therefore, the effects of fermentation on butanol production by the FW-G and FW-D groups were re-evaluated after adjusting the solid to liquid ratio to 1:2 (*w/v*) (Table 3). Following saccharification, the initial glucose concentration in the FW-G group was 54.3 g/L, whereas the initial glucose concentration in the un-saccharified FW-D group was only 4.88 g/L, although its initial starch concentration was higher (46.9 g/L). The results after 72 h of fermentation are presented in Figure 3. The fermentation trends in the two groups were similar. The substrate glucose was gradually consumed by strain N1-4, and the concentration of the product (total solvent and butanol) gradually increased. Among them, the FW-G group consumed 46.3 g/L of glucose, and the total solvent was 12.2 g/L, of which butanol was 9.72 g/L. After adjusting the solid–liquid ratio to 1:2 (*w/v*), substrate inhibition caused by high glucose concentrations following saccharification was avoided. Although FW-G and FW-D (solid to liquid ratio 1:2 [*w/v*]) butanol productions were the same, the butanol yield (0.363 C-mol/C-mol) in the FW-D group and the maximum butanol production rate (0.512 g/L/h) were 1.07 and 2.05 times more than those of the FW-G group, respectively (Table 3). This was due to the fact that the glucose concentration was always less than 5 g/L during the direct fermentation of FW (Figure 3b). Strain N1-4 can secrete amylase, which can decompose the substrate starch while using the decomposed substrate

to produce a solvent. Therefore, the glucose concentration in the system was maintained at a low level, which avoids substrate inhibition and improves fermentation efficiency.

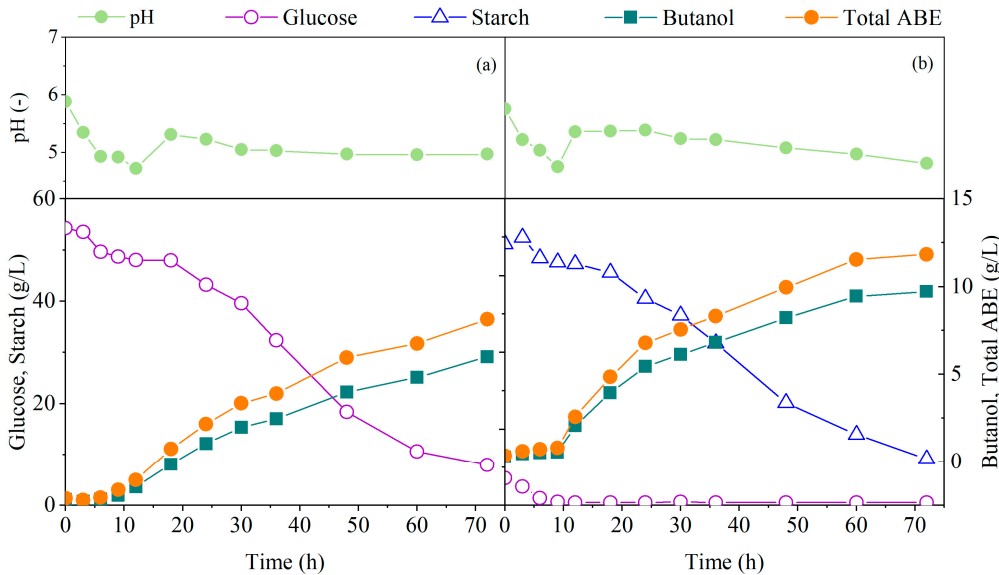

**Figure 3.** Time course of solvents produced from food waste with different saccharification treatments (solid–liquid ratio of 1:2 (*w/v*)). (**a**) Butanol production from food waste with saccharification; (**b**) butanol production from food waste without (direct) saccharification.

The abovementioned results show that it is not only feasible to ferment FW with direct strain N1-4 inoculation, but also that the butanol production efficiency and rate are better than those of the pre-saccharification process. Moreover, FW can be directly fermented at solid–liquid ratios of 1:2 (*w/v*) and 1:1 (*w/v*). The starch contained in the FW was saccharified and utilized by the N1-4 strain, avoiding substrate inhibition due to the high monosaccharide concentration.

### 3.3. Effects of Starch/Glucose Concentrations on Butanol Production

The abovementioned results show that the concentration of the carbon source has a significant effect on the production of butanol by the fermentation of strain N1-4. A suitable substrate concentration not only leads to a high level of fermentation efficiency, but also avoids wasting raw material [30]. Therefore, the effects of the concentration of glucose and starch substrates on ABE fermentation were investigated. A total of 40–80 g/L of glucose (G40, G50, G60, and G80) or starch (S40, S50, S60, and S80) were added to the TYA medium, and 10% (*v/v*) of strain N1-4 was inoculated. The solvent characteristics and fermentation kinetics parameters were compared during 72 h of fermentation at 30 °C.

In 40–80 g/L of glucose concentration, strain N1-4 could use the glucose substrate to produce solvents (Figure 4a–d). The G50 group presented the highest concentration of total solvent (16.4 g/L), butanol (12.1 g/L), and butanol productivity (0.679 g/L/h) following 12 h of fermentation. By increasing the initial glucose concentration to 60 and 80 g/L (G60, G80), the concentrations of butanol (11.4 g/L, 10.9 g/L) and total solvent (15.3 g/L, 14.7 g/L) did not increase; however, the maximum butanol productivity value (0.515 g/L/h, 0.382 g/L/h) decreased. In particular, the G80 group reached the maximum fermentation rate following 24 h of fermentation, which significantly lagged behind the other three groups (Table 4), and a large quantity of glucose was not consumed. These results indicate that a high, initial concentration of glucose (≥60 g/L) can inhibit the fermentation of strain N1-4. It was further determined that when the solid–liquid ratio was 1:1 (*w/v*) (experiment described in Section 3.2), the saccharification of FW converted all starch into glucose, and the excessively high glucose concentration (>100 g/L) led to fermentation failure (Figure 2b,c; Table 3). A previous study applied *C. beijerinckii* P260 for ABE fermentation

using wheat straw hydrolysate and similarly found that a high concentration of glucose (>62.5 g/L) resulted in significant substrate inhibition and a decreased solvent yield [7]. A high concentration of glucose affects the osmotic pressure of the cell owing to the size of the molecule. Following inoculation, the strain must adapt to an environment with a high osmotic pressure, leading to the prolongation of the lag phase during growth [31]. In this study, under 80 g/L glucose conditions, the lag phase of strain N1-4 was 24 h.

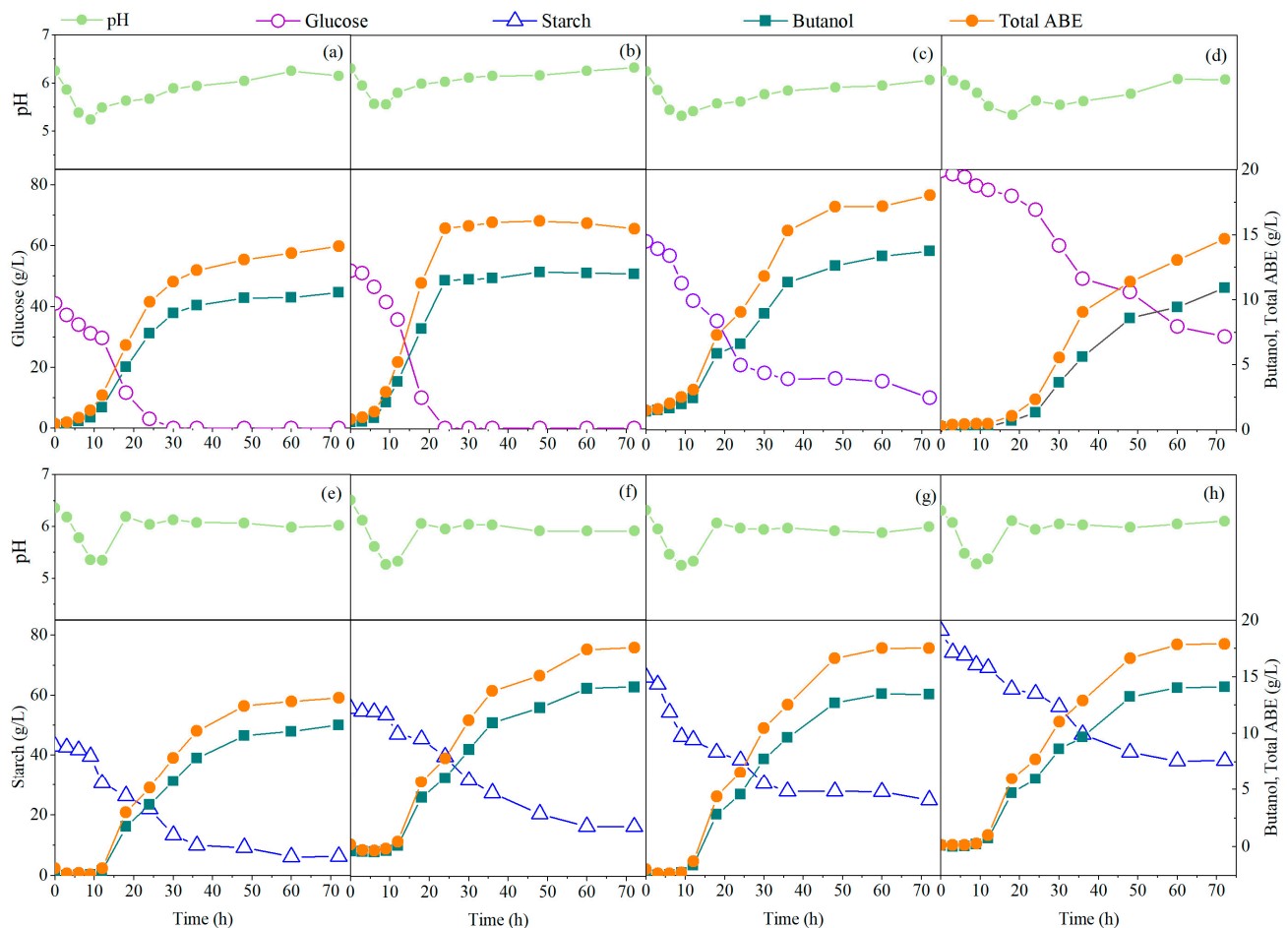

**Figure 4.** The influence of glucose and starch concentrations on ABE fermentation (**a–d**): concentrations of 40–80 g/L glucose (G40, G50, G60, G80); (**e–h**) concentrations of 40–80 g/L starch (S40, S50, S60, S80).

In contrast, with the increase in the starch concentration (40–60 g/L), the total solvent yield gradually increased (13.9–17.9 g/L). Butanol production did not continue to increase when the starch concentration exceeded 60 g/L, as butanol is toxic to microorganisms when it reaches a certain concentration [32]. However, fermentation was not significantly inhibited. S80 and S60 reached maximum butanol productivities of 0.655 and 0.653 g/L/h, respectively, following 12 h of fermentation (Figure 4e–h; Table 4). The S60 and S80 groups consumed 41.4 and 43.4 g/L of starch for 72 h, respectively, and a high amount of substrate residue remained. Therefore, the excessive, initial starch concentration created raw material waste. In addition, the butanol yield from starch was higher than that of glucose at the same concentration. When starch was used as the substrate, no evident inhibition was observed at a concentration of 80 g/L. That is, starch is a macromolecular polymer, and a high concentration of starch has less influence on osmotic pressure than glucose. However, studies have shown that high concentrations of sago [18] and cassava [33] starches affect ABE fermentation. The high concentration of starch increased the viscosity of the fermentation system and further affected the normal fermentation process.

**Table 4.** Effect of glucose/starch concentrations on solvents produced by strain N1-4.

| Carbon Source | Group [a] | Max. Solvents Production (g/L) | | | | Substrates Consumption (g/L) | Butanol Yield (C-mol/C-mol) | Maximum Butanol Production Rate [b] (g/L/h) |
|---|---|---|---|---|---|---|---|---|
| | | Acetone | Butanol | Ethanol | Total ABE | | | |
| Glucose | G40 | 2.95 | 10.5 | 0.647 | 14.1 | 41.0 | 0.407 | 0.519 (12−18 h) |
| | G50 | 3.81 | 12.1 | 0.483 | 16.4 | 51.6 | 0.364 | 0.679 (12−18 h) |
| | G60 | 2.80 | 11.4 | 1.08 | 15.3 | 51.2 | 0.351 | 0.515 (12−18 h) |
| | G80 | 2.77 | 10.9 | 1.01 | 14.7 | 54.1 | 0.320 | 0.382 (24−30 h) |
| Starch | S40 | 1.76 | 11.8 | 0.370 | 13.9 | 36.6 | 0.472 | 0.567 (12−18 h) |
| | S50 | 2.70 | 13.5 | 0.599 | 16.7 | 39.2 | 0.505 | 0.647 (12−18 h) |
| | S60 | 2.92 | 14.2 | 0.725 | 17.8 | 41.1 | 0.506 | 0.653 (12−18 h) |
| | S80 | 3.14 | 14.0 | 0.683 | 17.9 | 43.4 | 0.483 | 0.655 (12−18 h) |

Each batch culture was independently performed using *C. saccharoperbutylacetonicum* N1-4 at 30 °C for 72 h in TYA medium containing the indicated substrate (working volume: 80 mL). [a] G40, glucose 40 g/L; G50, glucose 50 g/L; G60, glucose 60 g/L; G80, glucose 80 g/L; S40, starch 40 g/L; S50, starch 50 g/L; S60, starch 60 g/L; S80, starch 80 g/L. [b] Butanol production rate (g/L/h) = $(C_2 - C_1)/(t_2 - t_1)$, where $C$ is the butanol concentration (g/L), $t$ is the sampling time (h); the period for the calculation is shown in the following parenthesis.

Furthermore, these results show that the starch substrate without saccharification can be directly used by strain N1-4, and the substrate inhibition effect of glucose on strain N1-4 is greater than that of starch. In other words, ABE fermentation can be directly conducted in the presence of high concentrations of starchy substrates, which can improve the solid–liquid ratio of the fermentation of raw materials and produce higher butanol production and carbon conversion rates, decreasing water usage and the cost of subsequent water treatment.

### 3.4. Acetone–Butanol–Ethanol Fermentation Characteristics of Starchy Substrate

Lignocellulosic materials composed of xylan, arabinan, and dextran are used as substrates for the fermentation of strain N1-4 [34]. Although studies have demonstrated that strain N1-4 has the ability to metabolize five (xylose and arabinose) and six (glucose) carbon sugars [20], the fermentation efficiency in the mixed state was reduced by as much as 50–70%, owing to carbon catabolite repression (CCR) [35]. In contrast, the FW used in this study was primarily composed of homogeneous glucose polymers (starch and dextran) with a concentration of 65.73% (Table 1). Therefore, fermentation with strain N4–1 was not inhibited by CCR, and the fermentation efficiency was high. In addition, the use of starch materials for butanol production requires the addition of extra nitrogen sources, buffer materials, and trace metals. The FW used in this study was rich in nutrients and had a buffering effect. The FW had a high carbon to nitrogen ratio (22.4) and contained abundant trace metals, such as sodium, potassium, magnesium, calcium, iron, manganese, zinc, copper, and selenium. The most abundant elements in the FW were calcium (4.95 mg/g), potassium (2.10 mg/g), sodium (0.694 mg/g), magnesium (0.490 mg/g), and zinc (0.148 mg/g) (Table 5). Calcium salt can be used as a buffer substance for fermentation systems due to its rich calcium content (insoluble calcium accounts for 91.6%) [33], accounting for the buffering capacity of the FW. When the solid to liquid ratio of the direct fermentation experiment (FW-D) decreased from 1:1 to 1:2; the calcium salt content, buffer capacity, and pH value of the system also decreased (Figures 2a and 3b). In addition, $Ca^{2+}$ can improve the stability of the cell membrane [36].

**Table 5.** Mineral composition of canteen food waste (FW).

| Elements | Na | K | Mg | Ca | Fe | Mn | Zn | Cu | Se |
|---|---|---|---|---|---|---|---|---|---|
| FW (mg/g) [a] | 0.694 | 2.10 | 0.490 | 4.95 | $46.0 \times 10^{-3}$ | $4.51 \times 10^{-3}$ | 0.148 | $7.59 \times 10^{-3}$ | $1.24 \times 10^{-3}$ |

[a] The parameters are calculated on dry matter.

There are numerous studies on ABE fermentation by inoculating different kinds of ABE-producing bacteria and using different starch materials (sago, corn, potatoes, cassava, oat, and waste obtained from the food processing industry) (Table 6). The production of butanol and total solvent was between 3.34–17.8 and 4.62–21.0 g/L, respectively, and the butanol yield was between 0.164–0.601 C-mol/C-mol, which depended on the initial starch content and species of strain. High butanol production rates (15.5–17.8 g/L) can be obtained by strain N1-4 using different types of pure starch substrates. The butanol yield of strain N1-4 using pure grain starch as the raw material was 0.535–0.601 C-mol/C-mol [17].

**Table 6.** Fermentation characteristics of starchy substrate using ABE-producing clostridia.

| Substrate | Starch Concentration (g/L) | Producer Strain | Nutrientsaddition [a] | Butanol (g/L) | ABE (g/L) | Butanol Yield (C-mol/C-mol) | Butanol Production Rate (g/L/h) | Reference |
|---|---|---|---|---|---|---|---|---|
| Sago starch | 60 | *C. acetobutylicum* P262 | Nitrogen | 16.0 | 18.0 | 0.390 | 0.290 [b] | [18] |
| Corn starch | 30 | *C. acetobutylicum* P262 | Nitrogen | 8.61 | 11.9 | 0.424 | 0.180 [b] | [18] |
| | 49.32 | *C. beijerinckii* NCIMB 8052 | P2 | 9.30 | 12.3 | — [c] | 0.378 [d] | [16] |
| | 58.1 | *C. saccharoperbutylacetonicum* N1-4 | TYA | 17.5 | 22.4 | 0.569 | 0.686 [d] | [17] |
| Potato starch | 30 | *C. acetobutylicum* P262 | Nitrogen | 3.34 | 4.62 | 0.164 | 0.06 [b] | [18] |
| | 110 | *C. beijerinckii* CCM 6218 | NO | 4.73 | 7.69 | — | — | [8] |
| | 110 | *C. saccharoperbutylacetonicum* DSM 14923 | NO | 3.83 | 6.18 | — | — | [8] |
| Cassava starch | 30 | *C. acetobutylicum* P262 | Nitrogen | 4.89 | 6.74 | 0.236 | 0.160 [b] | [18] |
| Wheat starch | 56.5 | *C. saccharoperbutylacetonicum* N1-4 | TYA | 17.8 | 22.0 | 0.601 | 0.692 [d] | [17] |
| Inedible dough | 49.91 | *C. beijerinckii* NCIMB 8052 | P2 | 9.30 | 14.4 | — | 0.412 [d] | [16] |
| Breading | 49.32 | *C. beijerinckii* NCIMB 8052 | P2 | 10.5 | 14.8 | — | 0.290 [d] | [16] |
| Batter liquid | 50.55 | *C. beijerinckii* NCIMB 8052 | P2 | 10.0 | 15.1 | — | 0.449 [d] | [16] |
| Modeling FW | 129 [e] | *C. beijerinckii* P260 | P2 | 13.2 | 19.7 | 0.390 | 0.560 [d] | [15] |
| | 181 [e] | *C. beijerinckii* P260 | P2 | 12.8 | 20.9 | 0.368 | 0.663 [d] | [15] |
| Canteen FW | 92.5 | *C. saccharoperbutylacetonicum* N1-4 | NO | 12.1 | 16.1 | 0.402 | 0.705 | This study |
| | 46.9 | *C. saccharoperbutylacetonicum* N1-4 | NO | 9.72 | 11.8 | 0.363 | 0.512 | This study |

[a] Nitrogen (nitrogen source): yeast extract (5 g/L), $NH_4NO_3$ (2 g/L); TYA (TYA stock): $CH_3COONH_4$ (30 g/L), $MgSO_4 \cdot 7H_2O$ (3 g/L), $KH_2PO_4$ (5 g/L), $FeSO_4 \cdot 7H_2O$ (100 mg/L), yeast extract (20 g/L) and tryptone (60 g/L); P2 (P2 stock): yeast extract (1 g/L), vitamin (0.1 g/L para-amino-benzoic acid, 0.1 g/L thiamine, 0.001 g/L biotin), buffer (50 g/L $KH_2PO_4$, 50 g/L $K_2HPO4$, 220 g/L $CH_3COONH_4$), mineral (20 g/L $MgSO_4.7H_2O$, 1 g/L $MnSO4.7H_2O$, 1 g/L $FeSO4.7H_2O$, 1 g/L NaCl). [b] Average butanol production rate. [c] Not mentioned. [d] The data were cited from the approximate values of figures in published papers. [e] The initial concentration of additional waste biomass.

In recent years, the development and utilization of cheap biomass as a raw fermentation material for butanol production has become a popular research topic; however, few

studies have reported the use of nutritious FW as a fermentation substrate. In this study, the strain N1-4 was used for the first time to ferment real FW with an initial starch content of 92.5 g/L. The butanol production in this study (12.1 g/L) was comparable to the highest butanol yield of other strains (3.83–13.2 g/L). Furthermore, the butanol production results in this study are similar to those reported by Huang et al. [15], who used simulated FW to produce butanol (0.402 C-mol/C-mol). In addition, the rate of butanol production in this study by strain N1-4, directly obtained from the FW, reached 0.705 g/h/L, which was similar to the previously reported values ranging from 0.648–0.692 g/h/L, and was markedly better than the related research reports for other strains (0.290–0.663 g/h/L).

## 4. Conclusions

Direct butanol production from starchy FW by strain N1-4 without saccharification and nutrient supplementation was investigated. Direct fermentation avoided substrate inhibition due to saccharification and produced 12.1 g/L of butanol at a production rate of 0.705 g/L/h and yield of 0.402 C-mol/C-mol with a solid–liquid ratio of 1:1 (*w/v*). The maximum butanol production rate in the direct mode was 2.05-fold higher than that in the saccharified mode (solid–liquid ratio of 1:2 (*w/v*)). Furthermore, FW was observed to be rich in trace elements and nutrients and could act as a low-cost and promising feedstock for efficient, direct, butanol production.

**Supplementary Materials:** The following supporting information can be downloaded at: https://www.mdpi.com/article/10.3390/su15076061/s1, Figure S1: Time course of batch culture with different ABE-producing clostridia using 50 g/L glucose as substrate. (a) *Clostridium saccharoperbutylacetonicum* N1-4; (b) *Clostridium acetobutylicum* ATCC 824; (c) *Clostridium beijerinckii* NCIMB 8052. □, glucose concentration; △, dry cell weight; ●, butanol concentration; ◆, total solvents concentration; dashed line, pH; Figure S2: The experimental phenomenon of amylose and amylopectin gelatinization after autoclave in the batch cultures (preparation of starchy carbon source in the text tube experiments); Table S1: Solvents production by different ABE-producing clostridia using glucose as substrate.

**Author Contributions:** Writing—original draft, visualization, data curation, conceptualization, X.W.; writing—original draft, methodology, visualization, H.S.; data curation, visualization, Y.W.; validation, F.W.; validation, W.Z.; supervision, conceptualization, writing—review and editing, C.W.; supervision, writing—review and editing, Q.W.; supervision, conceptualization, writing—review and editing M.G. All authors have read and agreed to the published version of the manuscript.

**Funding:** This work was supported by the National Natural Science Foundation of China (51708024), the National Key R&D Program of China (2019YFC1906302 and 2022YFE0105700), and the National Environmental and Energy Base for International Science and Technology Cooperation.

**Institutional Review Board Statement:** Not applicable.

**Informed Consent Statement:** Not applicable.

**Data Availability Statement:** Not applicable.

**Conflicts of Interest:** The authors declare no conflict of interest.

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
