# Peer review of "Feasibility of Efficient, Direct, Butanol Production from Food Waste without Nutrient Supplement by Clostridium saccharoperbutylacetonicum N1-4"

_sustainability, doi:10.3390/su15076061_

Round 1
Reviewer 1 Report
The study presented by Wang et al. evaluates the production efficiency of acetone, butanol and ethanol from food waste fermentation by different Clostridium strains. Among the results, I highlight the butanol production by C. saccharoperbutylacetonicum N1-4 and the characterization of trace elements, which allows a sustainable destination for food waste. I consider the manuscript able to be accepted for publication, after minor revisions.
Line 47: Change "by-products" to "by-products.
Lines 90-92: I suggest that this sentence be allocated in Session 2.4, along with mention of Table 1.
Line 192: Which detector is coupled to the HPLC?
Table 2 and lines 301-302: Were statistical tests used to prove the statistically significant difference? Mention and bring up the p-values.
References: Only 6 references cited were published in the last 5 years. I suggest revision.
Reviewer 2 Report
The manuscript by Wang et al studies an alternative way of producing butanol from food waste. The manuscript is well written and of high quality. However, prior to acceptance some issues have to be addressed in order to improve the overall quality of the work.
1) In the introduction sector, please be more specific at how higher energy butanol yields compared to ethanol. Please provide a number.
2) The overall methodology is too long. It would be nice to include a flowchart of the experimental procedure so the readers can understand more easily your work.
3) Please provide explanation regarding the conditions you tested (i.e. pH value, solid:liquid ratio etc.)
4) Statistical analysis of your results would help on identifying and validating any significant differences between the studied parameters and the obtained results. Please include it in your manuscript.
